# Establishment and validation of a prognostic nomogram for severe fever with thrombocytopenia syndrome: A retrospective observational study

**Kai Yang, Yu Wang, Jiepeng Huang, Lingyan Xiao, Dongyang Shi, Daguang Cui, Tongyue Du, Yishan Zheng** [ID]*

Department of Intensive Care Unit, The Second Hospital of Nanjing, Affiliated to Nanjing University of Chinese Medicine, Nanjing, Jiangsu, China

* fsyy01473@njucm.edu.cn

**Data Availability Statement:** All relevant data are within the paper and its Supporting Information file.

## Abstract

### Background

Several scoring systems have been proposed to predict the risk of death due to severe fever with thrombocytopenia syndrome (STFS), but they have limitations. We developed a new prognostic nomogram for STFS-related death and compared its performance with previous scoring systems and the Acute Physiology and Chronic Health Evaluation score (APACHE II Score).

### Methods

A total of 292 STFS patients were retrospectively enrolled between January 2016 and March 2023. Boruta's algorithm and backward stepwise regression were used to select variables for constructing the nomogram. Time-dependent receiver operating characteristic (ROC) curves and clinical decision curves were generated to compare the strengths of the nomogram with others.

### Results

Age, Sequential Organ Failure Assessment Score (SOFA score), state of consciousness, continuous renal replacement therapy (CRRT), and D-dimer were significantly correlated with mortality in both univariate and multivariate analyses (P<0.05). We developed a nomogram using these variables to predict mortality risk, which outperformed the SFTS and APACHE II scores (Training ROC: 0.929 vs. 0.848 vs. 0.792; Validation ROC: 0.938 vs. 0.839 vs. 0.851; P<0.001). In the validation set, the SFTS model achieved an accuracy of 76.14%, a sensitivity of 95.31%, a specificity of 25.00%, a precision of 77.22%, and an F1 score of 85.32%. The nomogram showed a superior performance with an accuracy of 86.36%, a precision of 88.24%, a recall of 93.75%, and an F1 score of 90.91%.

**Funding:** This work is funded by Nanjing Second Hospital Reserve Talent Program (0316301), Jiangsu Provincial Health and Health Commission 2021 Medical Research Projects (Jiangsu Health Science and Education (2021) No. 149), 2021 Nanjing Health Science and Technology Development Special Funds Project (YKK21121), Postgraduate Research & Practice Innovation Program of Jiangsu Province (SJCX23_0856) and Nanjing Infectious Disease Clinical Medical Center; Innovation center for infectious disease of Jiangsu Province(NO.CXZX202232). The funders had no role in study design, data collection and analysis, decision to publish, or preparation of the manuscript.

**Competing interests:** The authors have declared that no competing interests exist.

## Conclusion

Age, consciousness, SOFA Score, CRRT, and D-Dimer are independent risk factors for STFS-related death. The nomogram based on these factors has an excellent performance in predicting STFS-related death and is recommended for clinical practice.

## Introduction

Severe fever with thrombocytopenia syndrome (SFTS) is a lethal disease caused by severe fever with thrombocytopenia syndrome virus (SFTSV), a member in the Phenuiviridae family and the Bandavirus genus [1]. The virus is transmitted through tick bites or contact with infected body fluids [2–4]. First identified in China in 2009, SFTS has also been reported in Japan, Korea, Vietnam, Myanmar, Taiwan, and Thailand [5–8]. The mortality rate due to SFTS is as high as 30% [2]. The activity of ticks, one important vector of SFTSV, increases with temperature [9]. Thus, the prevalence of SFTS is expected to rise with global warming [10].

SFTS severity is associated with elevated levels of cytokines, such as interleukin-6 (IL-6), IL-8, IL-10, interferon-alpha (IFN-α), and interferon-gamma (IFN-γ), tumor necrosis factor-alpha (TNF-α), which are linked to disease progression [11, 12]. In Mainland China, SFTS mainly attacks middle-aged and elderly farmers in rural areas [13]. Currently, however, there are no specific treatments or licensed vaccines for SFTS. In recent years, prognostic indicators based on various risk factors related to SFTS have been explored. However, they have not yielded satisfactory outcomes [14–16]. Our previous all-factor study has shown that the prognosis of SFTS is associated with age, antihypertensive drug application and Acute Physiology and Chronic Health Evaluation II (APACHE II) score, and a SFTS scoring system has been developed [17]. This system has shown a strong ability to predict the risk of death.

In this prospective observational study, we expanded the sample size, and conducted Boruta algorithm, stepwise regression and multifactorial logistic regression analyses to explore more independent factors. Using these variables, a new nomogram was established, and its accuracy was compared with those of SFTS score and APACHE II score.

## Methods

### Study population

This study included a total of 292 patients admitted to The Second Hospital of Nanjing over a period from 01/06/2016 to 30/09/2023. Their clinical and laboratory data at admission were retrospectively analyzed. Patients with other virus infections or serious chronic diseases were excluded. SFTS patients were diagnosed based on the presence of acute fever (with a temperature of 38°C or higher) and platelet count $<100 \times 10^9$/L), with lab-confirmed SFTS virus (SFTSV) infection by qRT-PCR. We randomized the study population into a derivation set and a validation set (7:3). Their demographics and treatments were balanced between the two sets. The derivation set was divided into survival and death groups. The study was approved by The Second Hospital of Nanjing ethics committees (2023-LS-ky-023). The data were accessed for research purposes at 01/10/2023 to 01/11/2023. Authors did not have access to information that could identify individual participants during or after data collection.

### Data collection

In this retrospective study, the clinical and laboratory (blood routine, biochemistry, coagulation) data of the 292 patients diagnosed with SFTS on admission were collected from the

Second Hospital of Nanjing Case Data System. The patients were categorized into survival and death groups. Patients' characteristics, including demographics, medical history, exposure history, comorbidities (e.g., myocardial damage), symptoms, signs, laboratory findings, and treatments (e.g., vasopressors, noninvasive and invasive respiratory support, and renal replacement therapy and primary anti-infective regimen) were recorded. APACHE II score and SOFA score were calculated during the first 24 h after admission to the intensive care unit.

Supportive therapies, such as continuous renal replacement therapy, intermittent mandatory ventilation (IMV), vasoactive drugs, ribavirin, hormones, antifungal drugs, antibacterial drugs, were used. Hemofiltration dialysis was set as veno-venous continuous hemodiafiltration (vv-CHDF) at a dose of 25–30 mL/(kg/h). Sodium citrate and calcium gluconate were used for anticoagulation in vitro, with a course of 12 hours. The pH and electrolyte changes were monitored by arterial blood gas every 4 hours. Vasoactive drugs, including norepinephrine, epinephrine, and dobutamine at doses of 0.1–20 mg/(kg/min), were used to maintain average arterial pressure at 65 mmHg. The ventilator was set at modes of assisted/controlled ventilation (A/C), volume control synchronized intermittent mandatory ventilation (VC-SIMV), and pressure control synchronized intermittent mandatory ventilation (PC-SIMV), maintaining peak pressure 40 cm $H_2O$, platform pressure 30 cm $H_2O$, and driving pressure 15 cm $H_2O$ and maintaining $SpO_2$ 88%.

Pharmacological treatments, such as anti-bacterial and fungal infections, anti-inflammatory and immune-boosting treatments, were applied at the discretion of the doctor depending on the condition of the patient at admission and during hospital stay.

## Construction of the nomogram

Firstly, both univariate analysis and the Boruta algorithm were applied to screen significant factors. The factors simultaneously with $P<0.05$ in the univariate analysis and the non-rejection in the Boruta algorithm were included into the backward stepwise regression analysis to explore the independent factors. Collinearity between continuous variables was tested by the variance inflation factor (VIF), and an arithmetic square root of VIF $\leq 2$ was considered as non-collinearity. MeanDecreaseGini and MeanDecreaseAccuracy analyses were used to visualize the importance of each factor. Finally, the results of the backward stepwise regression analysis were subjected to the binary logistic regression analysis to explore the independent factors affecting the prognosis of SFTS. Based on them, a nomogram was obtained in the training set according to Occam's Law of Razor. The best model should be one that could achieve the best performance with the least variables [18].

## Validation of the nomogram

The performances of the nomogram, SFTS score and APACHE II score were compared in terms of area under curve (AUC), integrated discrimination improvement (IDI), decision curve analysis (DCA) in both the training and validation sets. In the validation set, the total accuracy (TA) and consistency (Kappa coefficient) of the nomogram and SFTS score were compared.

In both training and validation sets, the performance of the nomogram was evaluated by an area under the curve of the receiver operating characteristic (AUROC) and by calibration with bootstrap method with 1000 resampling. Hosmer and Lemeshow Goodness of Fit (GOF) tests were applied to model calibration evaluation. DCA analysis was performed to evaluate the net benefit of medical intervention conforming nomogram.

## Statistical analysis

Statistical analyses were performed using the R software (version4.3.1 R Foundation for Statistical Computing, Vienna, Austria). In the univariate analysis, categorical data were analyzed by $x^2$ test. Continuous data in a normal distribution were analyzed by independent t test and presented as $\bar{X} \pm s$. Continuous data in a non-normal distribution were analyzed by the Mann Whitney test and their medians were compared (P25, P75).

Missing values were addressed with multiple imputation in the process of logistic regression and model construction with five interpolations. The imputation technique involved creating multiple copies of the data and replacing missing values with imputed values through a suitable random sample from their predicted distribution. A two-tailed p value <0.05 was considered statistically significant. Candidate variables for multifactor analysis were screened using Brouta's algorithm in conjunction with univariate analysis, where green was significant, yellow was uncertain, and red was rejected in Brouta's algorithm visualization graph. Variables associated with death in the univariate analyses (p < 0.05) were included into the multivariate logistic regression analysis, and their estimated odds ratios (ORs) and 95% confidence intervals (95% CIs) were calculated, with a significance level of p < 0.05 for independent risk factors. We performed backward stepwise regression analysis on the dataset using the R package "stats". All analyses were reported according to the Transparent Reporting of a Multivariable Prediction Model for Individual Prognosis or Diagnosis (TRIPOD) guidelines [19].

## Results

### Patient description

Table 1 shows the baseline characteristics of patients at admission in the training and validation sets. The total patients were randomly grouped in a 7:3 ratio into the training and validation sets. Baseline characteristics data, except for Mean arterial pressure (MAP), were well balanced between the two cohorts. Between 2016 and 2023, a total of 302 patients had positive qRT-PCR results for SFTS virus. Cases with missing data or who died within 24 hours after admission were excluded. Then, 292 patients were finally involved (Fig 1), with a median age of 68 years (range 58–72 years). The demographic and pathological features of the patients are summarized in Table 1.

Table 2 shows that 48 of the 204 patients in the Derivation set died, with a mortality rate of 23.5%. The gender difference between the two groups was not significant. The median age of patients was 70.5 (64.75, 74.25) in the death group and 67 (56, 71.25) in the survival group. The demographic and pathological features in the training set are summarized in Table 2

### Development of a prediction nomogram in the training set

The Boruta algorithm was applied to select all the features associated with prognosis, and the Maximum Z Score among Shadowed Attributes (MZSA) was found to distinguish significant from non-significant features. A total of 12 variables (consciousness, APACHE Ⅱ Score, SOFA Score, heart rate (HR), vasopressors, medicine ventilator (MV), CRRT, blood urea nitrogen (BUN), aspartate transferase (AST), lactate dehydrogenase (LDH), creatinine (Cr), D-Dimer) were identified (green) as important features, three variables (antifungal, viral load, age) as pending, and the rest as rejected (red) (Fig 2A). The stabilization of variation of Z-scores during the Boruta run was shown (Fig 2B). Of the above 15 variables, 6 significant were filtered out in a backward stepwise regression analysis (age, consciousness, SOFA Score, vasopressors, CRRT, D-Dimer) and further subjected to a binary logistic regression analysis, further showing that age, consciousness, SOFA Score, vasopressors, CRRT, D-Dimer as independent risk

**Table 1. Demographic and pathological features of all the patients.**

| Variables | | Total (n = 292) | Derivation (n = 204) | Validation (n = 88) | P |
|---|---|---|---|---|---|
| Mortality | Non-survival | 72 (24.7%) | 48 (23.5%) | 24 (27.3%) | 0.594 |
| | Survival | 220 (75.3%) | 156 (76.5%) | 64 (72.7%) | |
| Sex | Male | 130 (44.5%) | 93 (45.6%) | 37 (42.0%) | 0.667 |
| | Female | 162 (55.5%) | 111 (54.4%) | 51 (58.0%) | |
| Age (years) | | 68 (58, 72) | 68 (57, 72.25) | 67 (59, 71) | 0.866 |
| Underlying diseases | Yes | 140 (47.9%) | 99 (48.5%) | 41 (46.6%) | 0.860 |
| | No | 152 (52.1%) | 105 (51.5%) | 47 (53.4%) | |
| Consciousness | Yes | 91 (31.2%) | 60 (29.4%) | 31 (35.2%) | 0.397 |
| | No | 201 (68.8%) | 144 (70.6%) | 57 (64.8%) | |
| APACHE II score | | 13 (9, 18) | 13 (9, 18) | 13 (8.75, 17) | 0.615 |
| SOFA score | | 3 (2, 5) | 3 (2, 5) | 3 (2, 5) | 0.564 |
| Viral load (copies/ml) | | 1.8 (0.08, 16) | 1.7 (0.11, 16) | 2.6 (0.07, 8.25) | 0.447 |
| IgM | Yes | 189 (64.7%) | 133 (65.2%) | 56 (63.6%) | 0.903 |
| | No | 103 (35.3%) | 71 (34.8%) | 32 (36.4%) | |
| IgG | Yes | 17 (5.8%) | 10 (4.9%) | 7 (8.0%) | 0.453 |
| | No | 275 (94.2%) | 194 (95.1%) | 81 (92.0%) | |
| T (°C) | | 38 (36.8,38.73) | 38.1(36.98,38.8) | 37.85 (36.8,38.6) | 0.288 |
| Heart Rate (Times/min) | | 84.5 (74, 94) | 85 (74, 94.25) | 83 (72, 93.25) | 0.946 |
| MAP (mmHg), Mean ± SD | | 83.4 ± 12.7 | 84.5 ± 12.2 | 81 ± 13.4 | 0.038* |
| Vasopressors | Yes | 20 (6.8%) | 15 (7.4%) | 5 (5.7%) | 0.79 |
| | No | 272 (93.2%) | 189 (92.6%) | 83 (94.3%) | |
| Ribavirin | Yes | 241 (82.5%) | 167 (81.9%) | 74 (84.1%) | 0.77 |
| | No | 51 (17.5) | 37 (18.1%) | 14 (15.9%) | |
| Favrovir | Yes | 39 (13.4%) | 28 (13.7%) | 11 (12.5%) | 0.924 |
| | No | 253 (86.6%) | 176 (86.3%) | 77 (87.5%) | |
| Immune_G | Yes | 174 (59.6%) | 126 (61.8%) | 48 (54.5%) | 0.306 |
| | No | 118 (40.4%) | 78 (38.2%) | 40 (45.5%) | |
| Hormone | Yes | 106 (36.3%) | 81 (39.7%) | 25 (28.4%) | 0.087 |
| | No | 186 (63.7%) | 123 (60.3%) | 63 (71.6%) | |
| Antifungal | Yes | 117 (40.1%) | 86 (42.2%) | 31 (35.2%) | 0.328 |
| | No | 175 (59.9%) | 118 (57.8%) | 57 (64.8%) | |
| Antibacterial | Yes | 244 (83.6%) | 168 (82.4%) | 76 (86.4%) | 0.499 |
| | No | 48 (16.4%) | 36 (17.6%) | 12 (13.6%) | |
| MV | Yes | 48 (16.4%) | 31 (15.2%) | 17 (19.3%) | 0.484 |
| | No | 244 (83.6%) | 173 (84.8%) | 71 (80.7%) | |
| CRRT | Yes | 43 (14.7%) | 29 (14.2%) | 20 (22.7%) | 0.106 |
| | No | 243(83.2%) | 175(85.8%) | 68 (77.3%) | |

APACHE II score = Acute Physiology and Chronic Health Evaluation score; SOFA score = Sequential Organ Failure Assessment Score; T(°C) = body temperature (°C); MAP = Mean Arterial Pressure; MV = Medicine Ventilator; CRRT = Continuous Renal Replacement Therapy. * P values compare the patient characteristics and outcome events in the development and validation cohorts using Wilcoxon Mann-Whitney test or exact Fisher test depending on whether the variable is continuous or categorical. $P < 0.05$ was considered significant and labeled with an asterisk (*) at the top corner of the P-value.

factors for poor prognosis in SFTS patients (Table 3). The VIF was calculated, and all variables in Fig 3A had an arithmetic square root of VIF ≤2. MeanDecreaseGini and MeanDecreaseAccuracy analyses indicated that the above five independent risks (Fig 3B) could be used as key variables for predicting the prognosis of patients with SFTS, and funneled into the nomogram scoring system (Fig 3C).

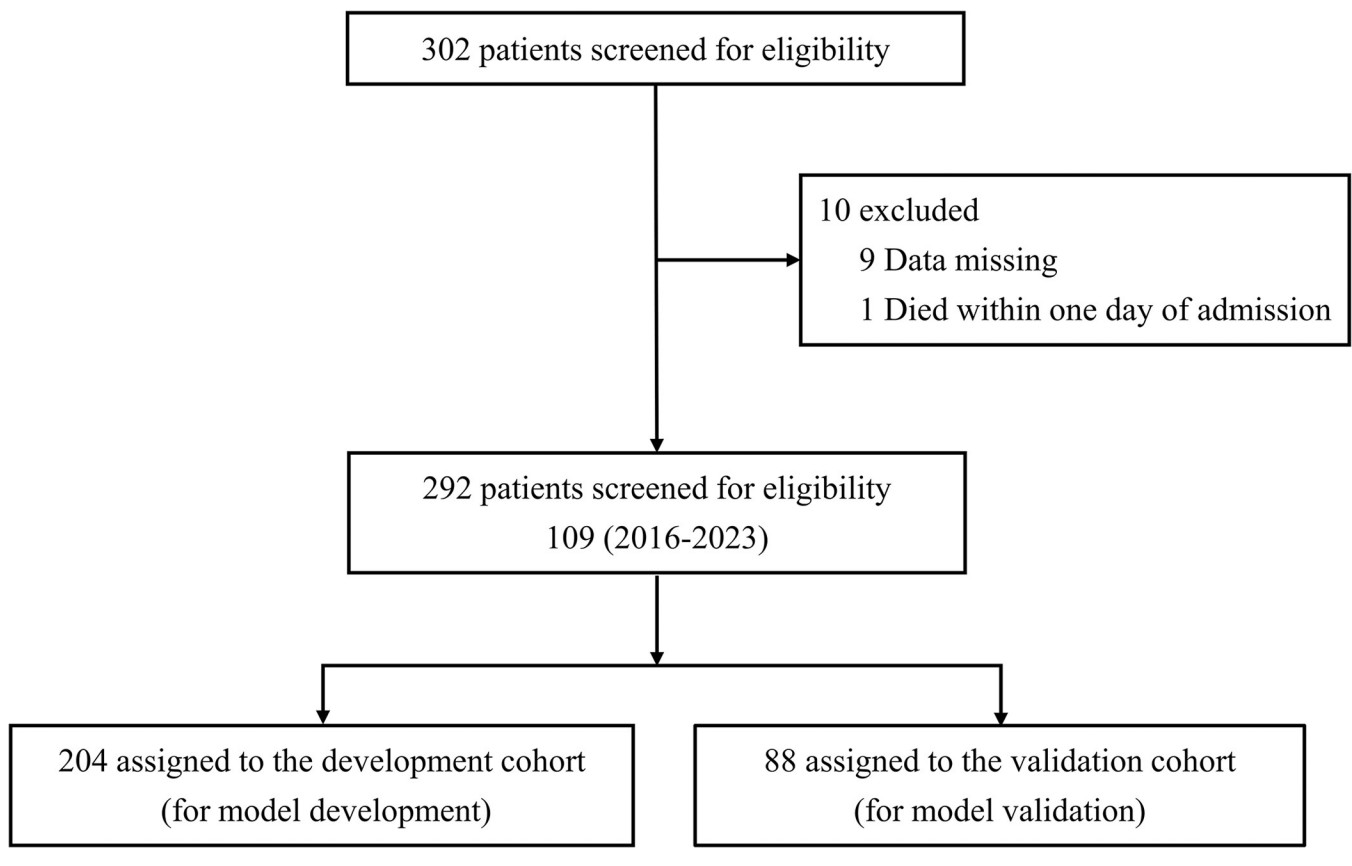

**Fig 1. Study flowchart.**

To apply the nomogram for prognosis prediction, patients' data, including age, consciousness, SOFA score, CRRT, and D-Dimer levels, were first collected. Following this, the total score was calculated by summing the points assigned to each of these factors according to the model. Finally, the total score was used to assess the patient's mortality risk, allowing for a nuanced evaluation of their prognosis.

## Comparison between three scoring systems

We compared the nomogram with SFTS score and APACHE II score in predicting the mortality in SFTS patients. As shown in Table 4, the AUROC of the nomogram was significantly higher than those of SFTS Score and APACHE II Score in both the training and validation sets, indicating that the nomogram had a stronger discriminative ability (Fig 4).

Calibration curves in both the training and validation sets revealed a good agreement between the actual and predicted probabilities (Fig 5). The values of Hosmer and Lemeshow Goodness of Fit (GOF) tests were 0.612 and 0.87 in the training and validation sets, respectively. The IDI of the nomogram scoring system was significantly higher than those of SFTS Score and APACHE II Score in both sets, revealing that the nomogram could increase the predictive accuracy of SFTS Score and APACHE II (Table 4). In the validation group, the accuracies of the nomogram and SFTS score were 86.4% (95% CI: 0.774, 0.928; $P = 0.002$) and 76.1% (95% CI: 0.659, 0.846; $P = 0.278$), respectively (Fig 6). The sensitivity and specificity of the nomogram were 93.8% and 66.7%, and those of SFTS score were 95.3% and 25.0%, respectively.

**Table 2. Patients' characteristics in the derivation set.**

| Variables | | Total (n = 204) | Survivor (n = 156) | Death (n = 48) | P |
|---|---|---|---|---|---|
| Sex | Male | 93 (45.6%) | 67 (42.9%) | 26 (54.2%) | 0.231 |
| | Female | 111 (54.4%) | 89 (57.1%) | 22 (45.8%) | |
| Age | | 68 (57, 72.25) | 67 (56, 71.25) | 70.5 (64.75, 74.25) | 0.005* |
| Underlying diseases | Yes | 99 (48.5%) | 69 (44.2%) | 30 (62.5%) | 0.04* |
| | No | 105 (51.5%) | 87 (55.8%) | 18 (37.5%) | |
| Consciousness | Yes | 60 (29.4%) | 22 (14.1%) | 38 (79.2%) | < 0.001* |
| | No | 144 (70.6%) | 134 (85.9%) | 10 (20.8%) | |
| APACHE II Score | | 13 (9, 18) | 11 (8, 16) | 19 (14, 23) | < 0.001* |
| SOFA Score | | 3 (2, 5) | 3 (2, 4) | 7 (5, 10) | < 0.001* |
| Viral load (copies/ml) | | 1.7 (0.11, 16) | 0.84 (0.06, 8.25) | 14.15 (1.41, 330) | < 0.001* |
| IgM | Yes | 133 (65.2%) | 103 (66.0%) | 30 (62.5%) | 0.783 |
| | No | 71 (34.8%) | 53 (34.0%) | 18 (37.5%) | |
| IgG | Yes | 10 (4.9%) | 8 (5.1%) | 2 (4.2%) | 1 |
| | No | 194 (95.1%) | 148 (94.9%) | 46 (95.8%) | |
| T(°C) | | 38.1 (36.98, 38.8) | 38 (36.8, 38.73) | 38.5 (37.5, 38.82) | 0.041* |
| HR (Times/min) | | 85 (74, 94.25) | 82 (68.75, 89.25) | 89.5 (83, 102.25) | < 0.001* |
| MAP, (mmHg) Mean ± SD | | 84.5 ± 12.2 | 84.6 ± 12.2 | 84 ± 12.3 | 0.772 |
| Vasopressors | Yes | 15 (7.4%) | 3 (1.9%) | 12 (25.0%) | < 0.001* |
| | No | 189 (92.6%) | 153 (98.1%) | 36 (75.0%) | |
| Ribavirin | Yes | 167 (81.9) | 126 (80.8) | 41 (85.4%) | 0.605 |
| | No | 37 (18.1) | 30 (19.2) | 7 (14.6%) | |
| Favrovir | Yes | 28 (13.7%) | 21 (13.5%) | 7 (14.6%) | 1 |
| | No | 176 (86.3%) | 135 (86.5%) | 41 (85.4%) | |
| Immune_G | Yes | 126 (61.8%) | 92 (59.0%) | 34 (70.8%) | 0.191 |
| | No | 78 (38.2%) | 64 (41.0%) | 14 (29.2%) | |
| Hormone | Yes | 81 (39.7%) | 51 (32.7%) | 30 (62.5%) | < 0.001* |
| | No | 123 (60.3%) | 105 (67.3%) | 18 (37.5%) | |
| Antifungal | Yes | 86 (42.2%) | 55 (35.3%) | 31 (64.6%) | < 0.001* |
| | No | 118 (57.8%) | 101 (64.7%) | 17 (35.4%) | |
| Antibacterial | Yes | 168 (82.4%) | 122 (78.2%) | 46 (95.8%) | 0.01* |
| | No | 36 (17.6%) | 34 (21.8%) | 2 (4.2%) | |
| MV | Yes | 31 (15.2%) | 8 (5.1%) | 23 (47.9%) | < 0.001* |
| | No | 173 (84.8%) | 148 (94.9%) | 25 (52.1%) | |
| CRRT | Yes | 29 (14.2%) | 9 (5.8%) | 20 (41.7%) | < 0.001* |
| | No | 175 (85.8%) | 147 (94.2%) | 28 (58.3%) | |
| WBC | | 3.34 (2.07, 5.71) | 3.48 (2.09, 6.06) | 3.13 (1.96, 4.48) | 0.298 |
| N($10^9$/L) | | 2.29 (1.15, 4.2) | 2.34 (1.15, 4.4) | 2.18 (1.14, 3.77) | 0.596 |
| PLT ($10^9$/L) | | 43.5 (32, 66) | 50 (34.75, 68.25) | 36 (26.75, 52) | < 0.001* |
| CRP (pg/mL) | | 1.12 (0.44, 7.18) | 1.31 (0.44, 8.16) | 1 (0.75, 4.97) | 0.704 |
| BUN (mmol/L) | | 5.74 (3.88, 8.09) | 5.04 (3.49, 7.48) | 7.77 (5.87, 11) | < 0.001* |
| TBil (mmol/L) | | 8 (5.9, 11.1) | 8 (5.8, 10.75) | 8.05 (6.77, 11.3) | 0.136 |
| ALT (U/L) | | 55.35 (34.9, 93.5) | 48.9 (32.73, 82.08) | 82.35 (49.73, 127.77) | < 0.001* |
| AST (U/L) | | 137.2 (71.57, 266.25) | 105.15 (58.65, 194.9) | 312.85 (166.88, 624.08) | < 0.001* |
| LDH (U/L) | | 598.5 (372.5, 998.25) | 503.5 (350.75, 825.75) | 1063.5 (705.5, 2091) | < 0.001* |
| ALP (U/L) | | 67 (49.75, 85) | 65 (48.92, 82.25) | 70 (57.75, 98.25) | 0.054 |

*(Continued)*

**Table 2.** (Continued)

| Variables | | Total (n = 204) | Survivor (n = 156) | Death | P |
|---|---|---|---|---|---|
| | | | | **(n = 48)** | |
| Cr (mmol/L) | | 75.45 (63, 91.6) | 72.4 (60.45, 84.12) | 90.5 (76.72, 125.25) | < 0.001* |
| D-Dimer | | 2.2 (0.95, 5.46) | 1.77 (0.82, 4.49) | 4.24 (1.94, 13.92) | < 0.001* |
| PNI | | 38.6 (35.34, 41.4) | 39.23 (36.18, 41.86) | 36.15 (33.24, 39.7) | 0.002* |

ALP = Alkaline Phosphatase; ALT = Alanine Transaminase; APACHE II score = Acute Physiology and Chronic Health Evaluation score; AST = Aspartate Transferase; BUN = Blood Urea Nitrogen; Cr = Creatinine; CRP = C-Reactive Protein; CRRT = Continuous Renal Replacement Therapy; HR = Heart Rate (times/min); LDH = Lactate Dehydrogenase; lgM = Immunoglobulin M; MAP = Mean Arterial Pressure; MV = Medicine Ventilator; N = Neutrophil count; PLT = platelets; PNI = Prognostic Nutritional Index; SOFA score = Sequential Organ Failure Assessment Score; T(˚C) = body temperature (˚C); TBil = Total Bilirubin; WBC = White Blood Cell. $P < 0.05$ was considered significant and labeled with an asterisk (*) at the top corner of the P-value.

The DCA for the nomogram was depicted and compared with those of SFTS score and APACHE II score. In the training set, medical intervention guided by the SFTS score could add more net benefit than the nomogram and APACHE II score, when the threshold probability (PT) was between 0.1 and 0.73. Nevertheless, the nomogram could add more net benefit

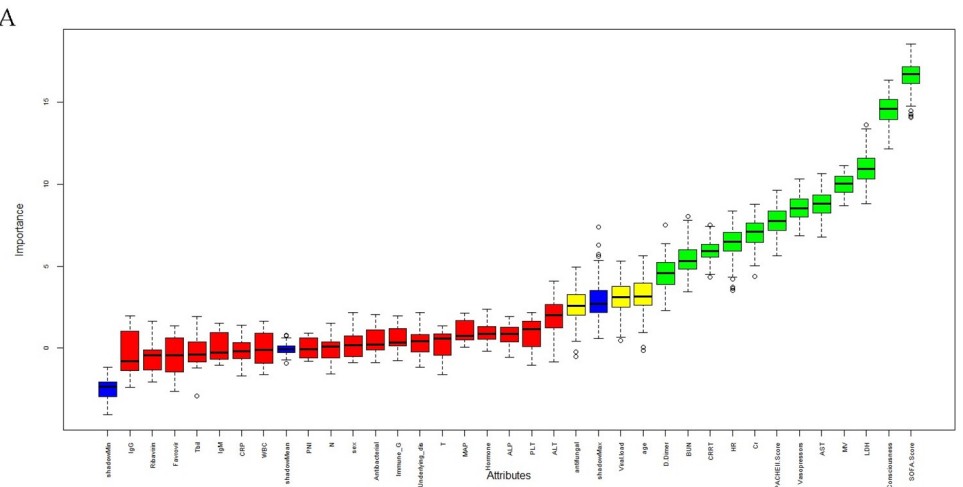

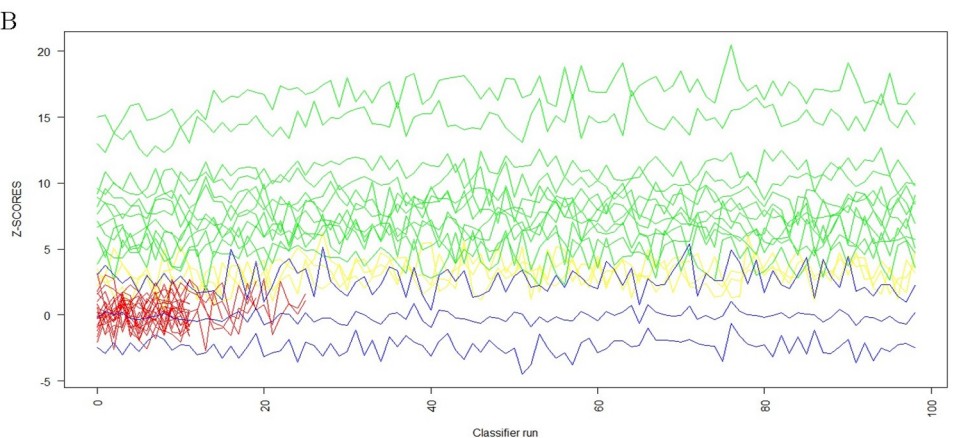

**Fig 2. Filtering of prognostic factors.** (A) Boruta variable selection graph; (B) Boruta iterative trajectory diagram.

**Table 3. Factors independently associated with death risk in SFTS patients in the multivariate logistic analysis.**

| Variables | OR | OR 95%CI | | P value |
|---|---|---|---|---|
| | | LL | UL | |
| Age | 1.11 | 1.04 | 1.18 | 0.002* |
| Consciousness | 4.48 | 1.42 | 14.14 | 0.011* |
| SOFA Score | 1.58 | 1.24 | 2.01 | 0.000* |
| Vasopressors | 4.71 | 0.67 | 33.29 | 0.120 |
| CRRT | 4.58 | 1.05 | 19.92 | 0.042* |
| D-Dimer | 1.05 | 1.01 | 1.09 | 0.006* |

CRRT = Continuous Renal Replacement Therapy; SOFA score = Sequential Organ Failure Assessment Score. $P<0.05$ was considered significant and labeled with an asterisk (*) at the top corner of the *P*-value.

than SFTS score and APACHE Ⅱ Score when the PT was between 0.73 and 0.1 (Fig 7A). In the validation set, treatment directed by SFTS score could gain more net benefit than the nomogram and APACHE Ⅱ Score, when the PT was between 0.1 and 0.6 (Fig 7B), but the nomogram could add more net benefit than the SFTS and APACHE Ⅱ Score when the PT was between 0.61 and 0.71(Fig 7B).

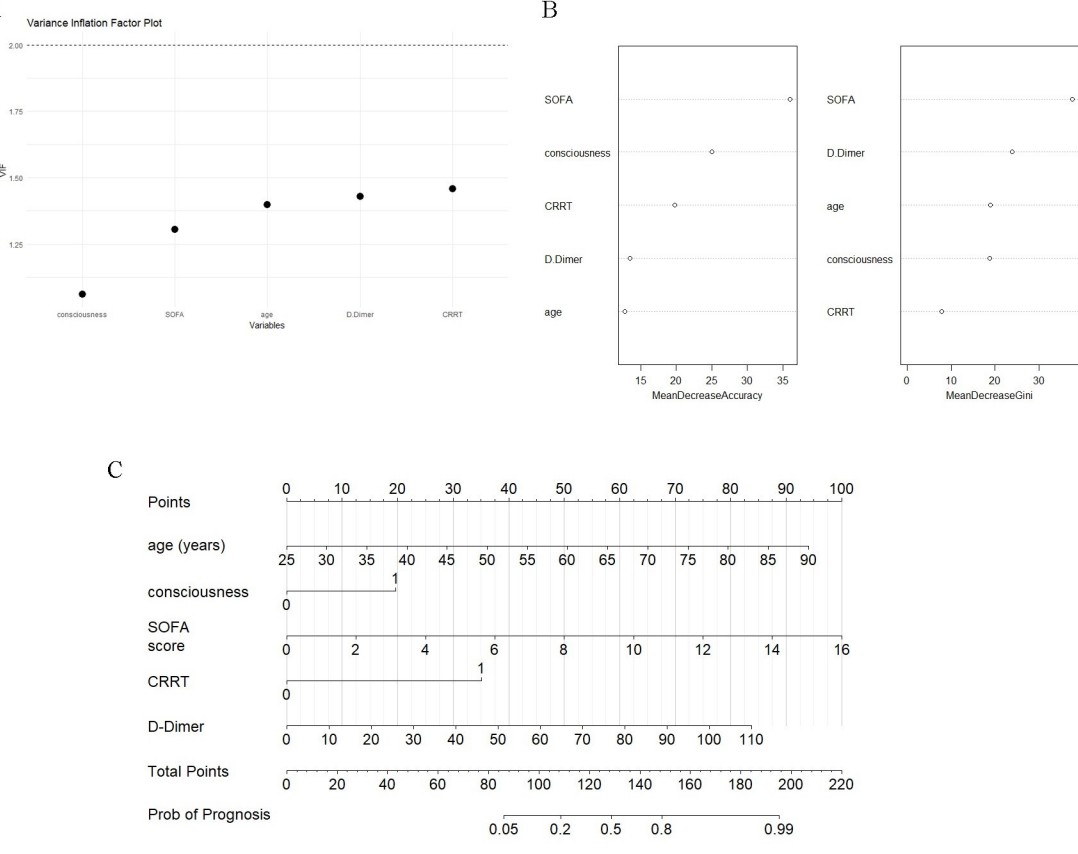

**Fig 3. Variable evaluation and construction of a prognostic nomogram.** (A) VIF values of less than 2 for each independent influencing factor; (B) MeanDecreaseGini and MeanDecreaseAccuracy analyses; (C) Nomogram scoring system.
CRRT = Continuous Renal Replacement Therapy; SOFA score = Sequential Organ Failure Assessment Score; VIF = Variance Inflation Factor.

Table 4. Comparison of models in predicting the mortality of SFTS.

| Predictive Model | | AUROC | *P value* | IDI | *P value* |
|---|---|---|---|---|---|
| Train set | Nomogram | 0.929 (0.883, 0.975) | | | |
| | SFTS Score | 0.848 (0.788, 0.908) | 0.012 | 0.281 (0.188, 0.379) | <0.001 |
| | APACHE II | 0.792 (0.722, 0.862) | 0.0002 | 0.344 (0.286, 0.461) | <0.001 |
| Test set | Nomogram | 0.938 (0.876, 1.001) | | | |
| | SFTS Score | 0.851 (0.765, 0.937) | 0.034 | 0.277 (0.161, 0.394) | <0.001 |
| | APACHE II | 0.839 (0.748, 0.929) | 0.008 | 0.295 (0.186, 0.404) | <0.001 |

APACHE II score = Acute Physiology and Chronic Health Evaluation score; AUROC = Area Under the curve of the Receiver Operating Characteristic; IDI = Integrated Discrimination Improvement; SFTS = Severe Fever with Thrombocytopenia Syndrome

## Discussion

By analyzing the clinical data of patients SFTS admitted at Nanjing Second Hospital (2016–2023), we conducted logistic regression analyses to recognize the risk factors related to mortality in STFS patients, including age, consciousness, SOFA score, CRRT and D-Dimer. Based on them, the nomogram achieved a satisfactory performance in predicting the mortality.

Nomograms allow researchers to visualize the prognostic outcomes of patients directly and quickly. Efficient nomograms need to be established on stable and significant variables, which are traditionally filtered out by a combination of single-factor and multi-factor analyses. However, these analyses have certain drawbacks, such as ignorance on interaction between factors, lack of a holistic view, existence of under-interpretation [15, 16]. Boruta's algorithm has high accuracy and robustness to identify important features by using shadow features and an iterative process [20]. Compared to other methods, Boruta's algorithm is superior in processing high-dimensional and nonlinear data, and can be integrated with other models [21]. We utilized the Boruta's algorithm (Fig 2) in conjunction with backward stepwise regression analysis

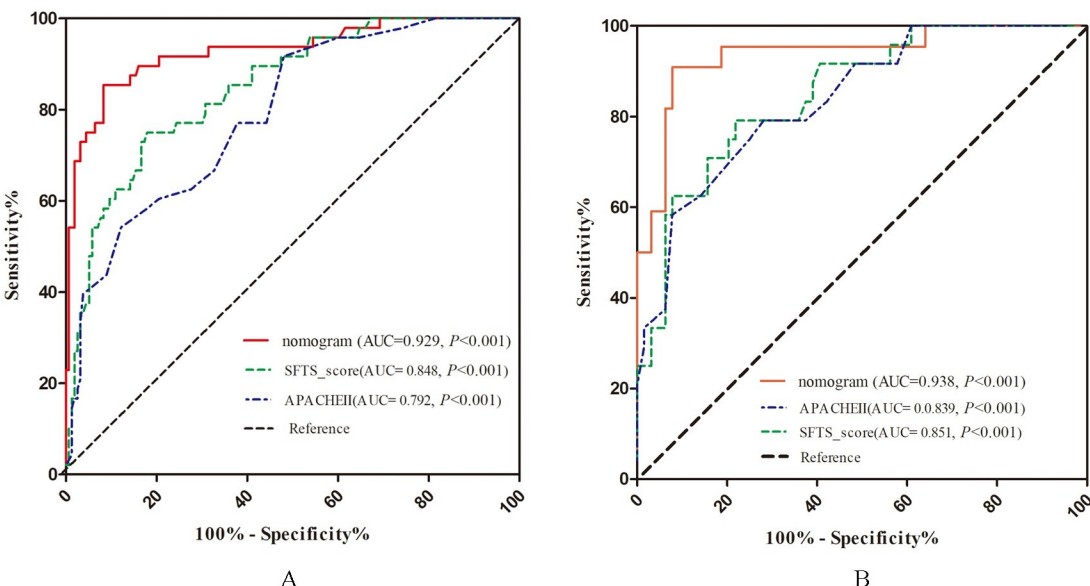

**Fig 4. Time-dependent ROC curves comparing the nomogram, SFTS score, and APACHEII score in predicting the death risk in patients with SFTS.** (A) Training set; (B) Validation set.

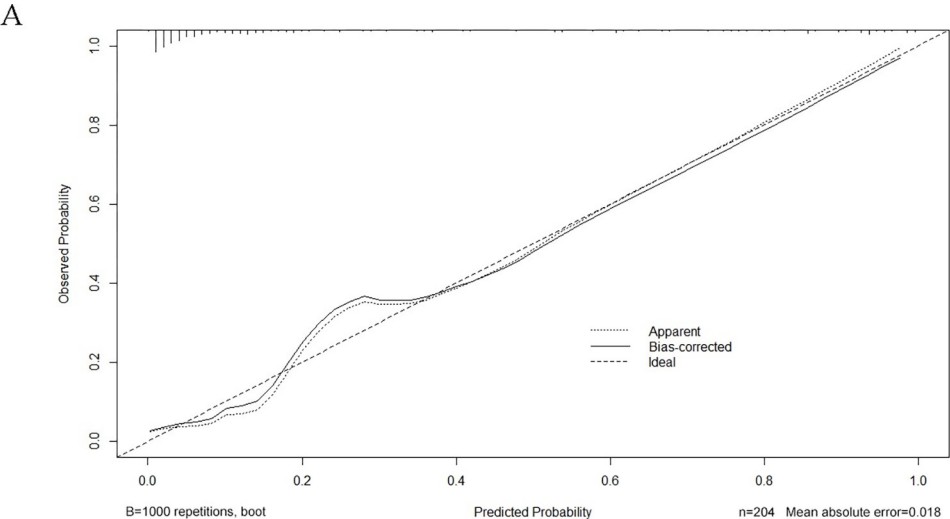

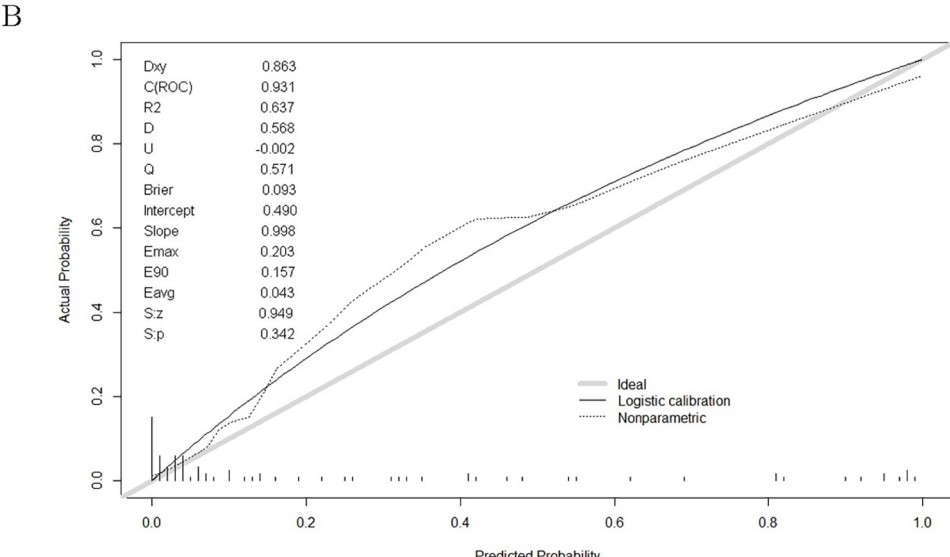

**Fig 5. Calibration plot of the nomogram predicting the death risk in patients with SFTS.** (A) Training set; (B) Validation set.

for feature selection on high-dimensional data, and screened out age, consciousness, SOFA Score, CRRT and D-Dimer (all with VIF<2) as independent factors (Fig 3A).

The present study showed that the mortality rate was as high as 24.7% in the total population, and the age of the patients in the death group was higher than that in the survivor group (Table 2). Age was an independent risk factor for prognosis (OR = 1.11, 95%CI: 1.04,1.18) (P = 0.002) (Table 3). There was no statistically significant difference between the two groups in terms of gender, which is consistent with the findings of Gong L, Yokomizo K and Chen Q, et al. [22–24]. An elder age indicated a higher mortality, which may be due to the fact that most elderly patients have a combination of underlying diseases, an increased risk of severe infections, a low organ compensatory capacity, and an impaired immunity [25, 26]. The SOFA

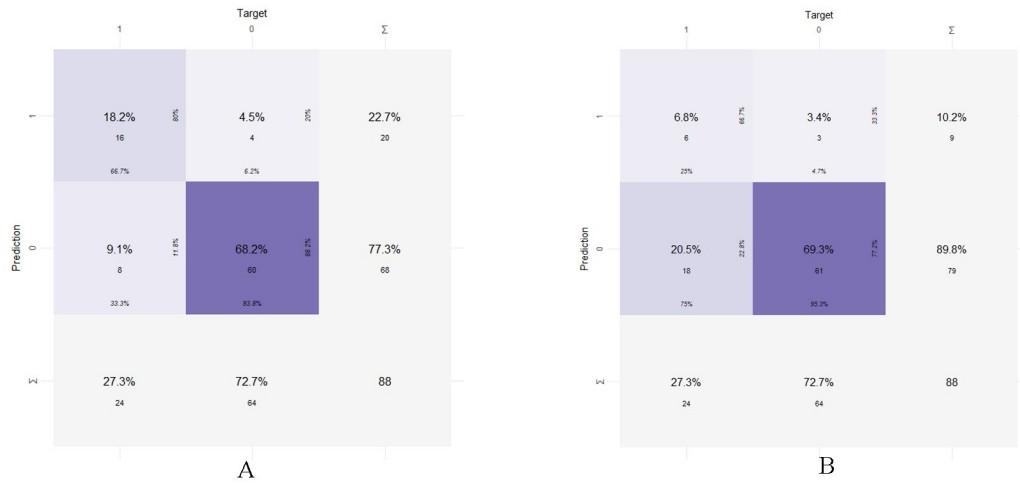

**Fig 6. Accuracy comparison between the nomogram and SFTS score in the validation set.** (A) Nomogram; (B) SFTS score.

score can be used to measure sepsis-related organ dysfunction in sepsis-related studies [27, 28], and perform well in assessing the severity of infectious diseases [29–31]. SFTV infection is mainly manifested as high fever, thrombocytopenia, leukocytopenia, gastrointestinal symptoms, liver and kidney function damage, and even MODS [32]. Karakike E et al. have noted that SOFA score has a high clinical value in predicting 28-day prognosis of infectious diseases [33]. Similarly, our study showed that the risk of death in patients with a high SOFA score was 1.58 times higher than that in patients with a low score, and our multifactorial analysis showed that SOFA score was an independent risk factor for the prognosis in patients with SFTS (OR = 1.58, 95% CI:1.24,2.01) ($P$<0.001) (Table 3). MeanDecreaseGini and MeanDecreaseAccuracy analyses indicated that SOFA score (Fig 3B) was the most important key variable in predicting the prognosis of patients with SFTS. Therefore, dynamic monitoring of SOFA score may be of great clinical value in assessing disease progression.

Consciousness is a component of the SOFA score. Studies have shown high morbidity and mortality in SFTS-related encephalitis or encephalopathy [34]. CNS manifestations (adjusted odds ratio [OR] 30·26) are important risk factors for the fatal outcomes of SFTS patients [35, 36]. Similarly, the present study showed that patients with altered consciousness had a 4.71 times higher risk of death than those without, indicating it as an independent prognostic factor for SFTS (OR = 4.71, 95% CI:1.42,14.14) ($P$ = 0.011) (Table 3). The pathogenic mechanism of encephalitis in patients with SFTS is unclear, and studies have shown that the virus is able to cross the blood-brain barrier into the skull to replicate, thus directly damaging neurons in brain tissue [37, 38]. D-dimer level has been the focus of studies about Coronavirus Disease 2019 (COVID-19) during the pandemic. Hypercoagulability may be a key mechanism for acute organ injury and death in COVID-19 patients. A higher D-dimer level has been found independently associated with a higher risk of death in a large multicenter cohort study of critically ill patients with COVID-2019 [39]. Meanwhile, coagulation abnormalities are also common in patients with STFS. Studies have shown that coagulation parameters are more sensitive than platelet count in early prediction of SFTS, with D-dimer levels much higher in the death group than in the survival group [40]. Similarly, the present study showed that the platelet count was lower in the death group than in the survival group at admission ($P$<0.001), but not an independent predictive factor for mortality; whereas, the D-dimer level (4.24 [1.94,13.92]) was higher than that in the survival group (1.77[0.82,4.49]) ($P$< 0.001), and was identified as

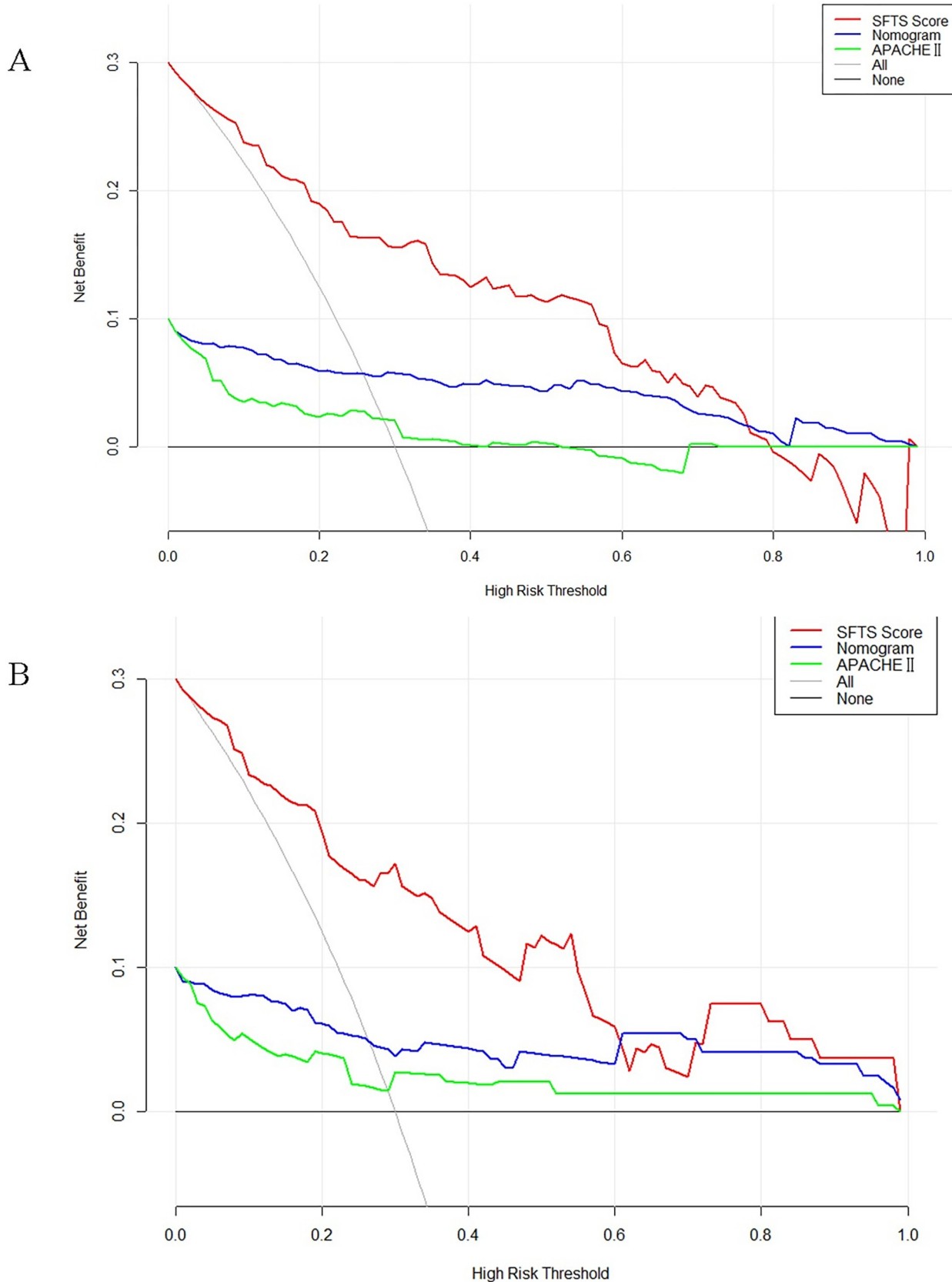

**Fig 7. Decision curve comparison among the nomogram, SFTS score, and APACHE II score in predicting the death risk in patients with SFTS.** (A) Training set; (B) Validation set.

an independent risk factor (OR = 1.05, 95%CI:1.01,1.09) (*P* = 0.006) in the multifactorial analysis (Table 3). Renal damage is one of the MOFs. Studies have shown that AKI is associated with poor outcomes in patients with SFTS, especially a high mortality in patients with AKI stage 2 or 3 [41]. In this study, we showed that the mortality rate was higher in patients who required CRRT at admission than those who did not, and the multifactorial analysis showed that the mortality was statistically 4.58 times higher in patients who required CRRT treatment than those who did not (*P* = 0.042) (Tables 2, 3).

Based on the five independent risk factors mentioned above, we constructed a prediction nomogram (Fig 3C), which achieved an accuracy of 86.36% in the validation set (Fig 6A). Currently, several scoring systems, such as APACHE score, SOFA score, and Simplified Acute Physiology Score II, have been proposed for prediction in ICU patients. APACHE II is the most widely used. Here, we further showed that the AUROC of the nomogram was 0.929 in the training set and 0.938 in the validation set, both higher than those of APACHE II score and the STFS score in our previous study (*P*<0.05), suggestive of its excellent discriminative ability (Fig 4A and 4B). In terms of IDI, the nomogram outperformed APACHE II score and STFS score (Table 4). The APACHE II score performed the worst, indicating that it may be not suitable to predict multiple organ failure [42].

This study has several advantages compared with the previous. This study validated that SFTS score system also had a good predictive ability and a high accuracy (training set: AUC = 0.848, validation set: AUC = 0.851) (Fig 4B), and surprisingly, the net clinical benefit of SFTS score was within a larger threshold range than the nomogram and APACHE II score (Fig 7A and 7B), and remained consistent across the training and validation sets. The net clinical benefit of the nomogram was higher than that of SFTS score within a certain threshold range. Therefore, we propose that both scoring systems may be complementary in clinical prediction of disease outcomes.

This study is limited by a single-center sample, a small validation set, and an insufficient generality. Due to the small sample size, the model may only be able to learn specific features in the training set and miss other meaningful features. Second, the subjects of the retrospective study may not be fully representative, and unidentified confounding factors may have biased the results.

In conclusion, the nomogram based on five factors (age, consciousness, SOFA score, CRRT and D-Dimer) showed an excellent performance in predicting the prognosis of STFS patients. The STFS score system, developed by us in the previous studies, has also a good predictive ability. Both are expected to be used in more clinical practices.

## Supporting information

**S1 File.**
(CSV)

## Author Contributions

**Conceptualization:** Kai Yang, Yishan Zheng.

**Data curation:** Kai Yang, Yu Wang, Jiepeng Huang.

**Funding acquisition:** Kai Yang, Yishan Zheng.

**Investigation:** Jiepeng Huang, Lingyan Xiao.

**Methodology:** Yu Wang, Lingyan Xiao, Dongyang Shi, Daguang Cui, Tongyue Du, Yishan Zheng.

**Project administration:** Kai Yang.

**Supervision:** Yishan Zheng.

**Validation:** Yu Wang.

**Visualization:** Dongyang Shi, Daguang Cui, Tongyue Du.

**Writing – original draft:** Kai Yang, Yu Wang, Jiepeng Huang, Lingyan Xiao.

**Writing – review & editing:** Kai Yang, Yishan Zheng.

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
