## [Decision Letter · Decision Letter 0]

24 Jul 2024

PONE-D-24-19302Establishment and validation of a prognostic nomogram for severe fever with thrombocytopenia syndrome: a retrospective observational studyPLOS ONE

Dear Dr. Zheng,

Thank you for submitting your manuscript to PLOS ONE. After careful consideration, we feel that it has merit but does not fully meet PLOS ONE’s publication criteria as it currently stands. Therefore, we invite you to submit a revised version of the manuscript that addresses the points raised during the review process.

We look forward to receiving your revised manuscript.

Kind regards,

Elvan Wiyarta, M.D.

Academic Editor

PLOS ONE

Journal Requirements:

3. We note that your Data Availability Statement is currently as follows: "All relevant data are within the manuscript and its Supporting Information files."

Reviewers' comments:

Reviewer's Responses to Questions

**Comments to the Author**

1. Is the manuscript technically sound, and do the data support the conclusions?

Reviewer #1: Yes

Reviewer #2: Yes

2. Has the statistical analysis been performed appropriately and rigorously? 

Reviewer #1: Yes

Reviewer #2: Yes

3. Have the authors made all data underlying the findings in their manuscript fully available?

Reviewer #1: Yes

Reviewer #2: Yes

4. Is the manuscript presented in an intelligible fashion and written in standard English?

Reviewer #1: Yes

Reviewer #2: Yes

5. Review Comments to the Author

Reviewer #1: This study developed and validated a nomogram prediction model based on age, SOFA score, consciousness status, CRRT, and D-dimer to predict the mortality risk of patients with severe fever with thrombocytopenia syndrome (SFTS), demonstrating superior performance compared to existing scoring systems. The manuscript has a clear overall structure, rigorous research methods, detailed data analysis, and reasonable interpretation of results. However, to further enhance the quality and readability of the manuscript, I propose the following revisions:

1. Some parts of the manuscript exhibit a somewhat Chinese style of expression. It is recommended to polish the language throughout the manuscript to ensure correct grammar, clear expression, and compliance with academic writing standards.

2. In the results section of the abstract, in addition to the ROC value, other performance indicators (such as sensitivity, specificity, accuracy, etc.) can be mentioned to comprehensively demonstrate the superiority of the new nomogram in this study.

3. It is suggested to add a brief explanation of the nomogram prediction model in the results section, including how to use the model specifically, to help readers better understand the necessity of the study and verify its clinical practicality.

4. Some details need modification, such as inconsistent capitalization of variables in Table 1.

Reviewer #2: Hyperproduction of several cytokines such as interleukin-6 (IL-6), IL-8, IL-10, IL-1β, tumour necrosis factor (TNF-α) and Interferon alpha (IFN-α), IFN-γ, and transforming growth factor‐β (TGF‐β) have been proved to correlate with fatal viral diseases such as SFTS patients.

: Could authors show cytokines data in this study?

In line 38, SFTSV virus

: Could the authors check this? SFTS virus or Severe Fever with Thrombocytopenia Syndrome Virus (SFTSV)

In line 43, the prevalence of SFTS is expected to rise with global warming.

: Could the authors put reference (s)?

In line 49, APACHE II

: Could the authors write full name of APACHE II?

In line 292, “the focus of studies about COVID-19 during the pandemic”

: Could authors also write full name of COVID19?

6. PLOS authors have the option to publish the peer review history of their article (what does this mean?). If published, this will include your full peer review and any attached files.

Reviewer #1: No

Reviewer #2: No

---

## [Author Response · Author response to Decision Letter 0]

14 Aug 2024

Dear Dr. Wiyarta and Reviewers,

We would like to express our sincere appreciation for your thorough review and constructive comments on our manuscript titled “Establishment and validation of a prognostic nomogram for severe fever with thrombocytopenia syndrome: a retrospective observational study” We have carefully considered each point raised and have made the necessary revisions. Below, we provide detailed responses to each comment and outline the changes made in the manuscript.

Response to Editor's Comments

Comment 1: Please ensure your manuscript complies with PLOS ONE's style requirements, including file naming conventions.

Response 1: We have reviewed and updated our manuscript to align with the PLOS ONE template and style guidelines.

Comment 2: Code and Data Availability Statement.

Response 2: We have revised the data availability statement in our submission to ensure it meets PLOS ONE's requirements.

Comment 3: ORCID ID

Response 3: We have added the ORCID ID for the corresponding author as requested.

Response to Reviewer #1

Comment 1: Some parts of the manuscript exhibit a somewhat Chinese style of expression. It is recommended to polish the language throughout the manuscript to ensure correct grammar, clear expression, and compliance with academic writing standards.

Response 1: We have polished the language throughout the manuscript to enhance clarity, correct grammar, and ensure it meets academic writing standards.

Comment 2: In the results section of the abstract, in addition to the ROC value, other performance indicators (such as sensitivity, specificity, accuracy, etc.) can be mentioned to comprehensively demonstrate the superiority of the new nomogram in this study.

Response 2: We have included additional performance indicators such as accuracy, sensitivity, specificity, precision, and F1 score in the results section of the abstract to provide a comprehensive demonstration of the model's performance.

Comment 3: It is suggested to add a brief explanation of the nomogram prediction model in the results section, including how to use the model specifically, to help readers better understand the necessity of the study and verify its clinical practicality.

Response 3: We have added a brief explanation of how to use the nomogram prediction model in the results section (line 200).

Comment 4: Some details need modification, such as inconsistent capitalization of variables in Table 1.

Response 4: The capitalization of variables in Table 1 has been standardized and corrected.

Response to Reviewer #2

Comment 1: Hyperproduction of several cytokines such as interleukin-6 (IL-6), IL-8, IL-10, IL-1β, tumor necrosis factor (TNF-α), Interferon alpha (IFN-α), IFN-γ, and transforming growth factor‐β (TGF‐β) have been proved to correlate with fatal viral diseases such as SFTS patients. Could authors show cytokine data in this study?

Response 1: In the introduction section, we discussed the association between cytokine levels and the severity of SFTS. However, cytokine data were not collected as part of this study. We acknowledge their potential importance in understanding disease mechanisms and prognosis. Future studies may explore these immunological factors to further refine predictive models and improve patient outcomes. Thank you for your interest in our research, and we look forward to addressing this aspect in future investigations.

Comment 2: 1) In line 38, SFTSV virus: Could the authors check this? SFTS virus or Severe Fever with Thrombocytopenia Syndrome Virus (SFTSV); 2) In line 43, the prevalence of SFTS is expected to rise with global warming: Could the authors put reference(s)?; 3) In line 49, APACHE II: Could the authors write the full name of APACHE II?; 4) In line 292, "the focus of studies about COVID-19 during the pandemic": Could authors also write the full name of COVID-19?

Response 2:

1) In line 42, the term "SFTSV virus" has been corrected to "Severe fever with thrombocytopenia syndrome virus (SFTSV)".

2) A relevant reference has been added to support the statement in line 48 regarding the expected increase in SFTS prevalence due to global warming.

3) The full name of APACHE II, "Acute Physiology and Chronic Health Evaluation II," has been provided (line 57).

4) The full name of COVID-19, "Coronavirus Disease 2019," has been included (line 305).

We believe these revisions and responses adequately address the comments and suggestions provided. We appreciate your consideration and look forward to your feedback.

Sincerely,

Yishan Zheng

---

## [Decision Letter · Decision Letter 1]

27 Sep 2024

Establishment and validation of a prognostic nomogram for severe fever with thrombocytopenia syndrome: a retrospective observational study

PONE-D-24-19302R1

Dear Dr. Zheng,

We’re pleased to inform you that your manuscript has been judged scientifically suitable for publication and will be formally accepted for publication once it meets all outstanding technical requirements.

Kind regards,

Elvan Wiyarta, M.D.

Academic Editor

PLOS ONE

Additional Editor Comments (optional):

Reviewers' comments:

Reviewer's Responses to Questions

**Comments to the Author**

1. If the authors have adequately addressed your comments raised in a previous round of review and you feel that this manuscript is now acceptable for publication, you may indicate that here to bypass the “Comments to the Author” section, enter your conflict of interest statement in the “Confidential to Editor” section, and submit your "Accept" recommendation.

Reviewer #1: All comments have been addressed

Reviewer #2: All comments have been addressed

2. Is the manuscript technically sound, and do the data support the conclusions?

Reviewer #1: Yes

Reviewer #2: Yes

3. Has the statistical analysis been performed appropriately and rigorously? 

Reviewer #1: Yes

Reviewer #2: Yes

4. Have the authors made all data underlying the findings in their manuscript fully available?

Reviewer #1: Yes

Reviewer #2: Yes

5. Is the manuscript presented in an intelligible fashion and written in standard English?

Reviewer #1: Yes

Reviewer #2: Yes

6. Review Comments to the Author

Reviewer #1: (No Response)

Reviewer #2: I have no additional comments for the author, including concerns about dual publication, research ethics, or publication ethics.

7. PLOS authors have the option to publish the peer review history of their article (what does this mean?). If published, this will include your full peer review and any attached files.

Reviewer #1: No

Reviewer #2: No

---

## [Editor Report · Acceptance letter]

9 Oct 2024

PONE-D-24-19302R1 

PLOS ONE

Dear Dr. Zheng, 

I'm pleased to inform you that your manuscript has been deemed suitable for publication in PLOS ONE. Congratulations! Your manuscript is now being handed over to our production team.

Kind regards, 

on behalf of

Mr. Elvan Wiyarta 

Academic Editor

PLOS ONE